# Statin Use Decreases the Incidence of Hepatocellular Carcinoma: An Updated Meta-Analysis

**DOI:** 10.3390/cancers12040874

**Published:** 2020-04-03

**Authors:** Antonio Facciorusso, Mohamed A Abd El Aziz, Siddharth Singh, Sara Pusceddu, Massimo Milione, Luca Giacomelli, Rodolfo Sacco

**Affiliations:** 1Section of Gastroenterology, Department of Medical Sciences, University of Foggia, 71122 Foggia, Italy; antonio.facciorusso@virgilio.it; 2Department of Surgery, Mayo Clinic, Rochester, MN 55905, USA; abdelmaksoud.mohamed@mayo.edu; 3Division of Gastroenterology, University of California, San Diego, CA 92093, USA; sis040@health.ucsd.edu; 4Fondazione IRCCS—Istituto Nazionale dei Tumori Via G. Venezian 1 IT, 20133 Milan, Italy; sara.pusceddu@istitutotumori.mi.it (S.P.); massimo.milione@istitutotumori.mi.it (M.M.); 5Department of Surgical Sciences and Integrated Diagnostics, University of Genoa, 16126 Genoa, Italy; luca.giacomelli@polistudium.it; 6Polistudium SRL, 20124 Milan, Italy

**Keywords:** cirrhosis, HCC, cancer, hazard ratio, survival

## Abstract

Statins can decrease hepatocellular carcinoma (HCC) occurrence, but the magnitude and the predictors of these effects remain unclear. This meta-analysis provides a pooled estimate of the impact of statin use on HCC occurrence. Pooled effects were calculated using a random-effects model by means of the DerSimonian and Laird test. Primary endpoint was the time-dependent correlation between statin use and HCC incidence expressed as hazard ratio (HR), both crude and adjusted. The crude and adjusted odds ratios (OR) for HCC occurrence between statin users and non-users were analyzed. Twenty-five studies with 1,925,964 patients were included. Crude OR for HCC incidence was 0.59 (95% CI: 0.47–0.74), confirmed in adjusted analysis (OR: 0.74, 95% CI: 0.70–0.78). Adjusted HR was 0.73 (95% CI: 0.69–0.76). This effect was more pronounced in HBV patients (HR: 0.46, 95% CI: 0.36–0.60) and with a cumulative daily dose beyond 365 (HR: 0.27, 95% CI: 0.11–0.67). Lipophilic statins were associated with reduced HCC incidence (HR: 0.49, 95% CI: 0.39–0.62). Atorvastatin determined the greater magnitude of effect (HR: 0.43, 95% CI: 0.28–0.65). This meta-analysis demonstrates the beneficial chemopreventive effect of statins against HCC occurrence. This effect is dose-dependent and more pronounced with lipophilic statins.

## 1. Introduction

Hepatocellular carcinoma (HCC) is the fifth most commonly occurring type of cancer and the leading cause of mortality in cirrhotic patients [1]. Despite the recent improvement in diagnosis and screening programs in cirrhotics, a great number of patients are still diagnosed in the advanced stage, thus being unsuitable to curative treatments, such as surgery, orthotopic liver transplantation, or radiofrequency ablation [2,3,4]. Therefore, the identification of prognostic factors for HCC occurrence is of paramount importance in order to decrease the burden of this disease, in particular in high-risk populations, such as cirrhotic or chronic hepatitis B virus (HBV) patients.

Three-hydroxy-3-methyl-glutaryl-coenzyme A (HMG co-A) reductase inhibitors (statins) are effective and commonly used worldwide as a treatment for dyslipidemia, and increasing evidence shows that statins also have anti-inflammatory and anti-oncogenic effects [5,6].

A previous meta-analysis, based on 10 studies, shed light on the beneficial role of statins in preventing HCC occurrence [7]; however, the relatively low number of studies included did not allow subgroup analyses based on the specific agent administered (type of statin) or to correlate the anticancer effect with the dose.

In the last years, several cohort and case–control studies have been published in the field, hence the need to update the previous data in order to better define the eventual chemopreventive role of statins in hepato-oncology.

The aim of this meta-analysis was to provide a pooled estimate of the impact of statin use on HCC occurrence. Primary endpoint was defined as the time-dependent correlation between statin use and HCC incidence (in terms of both crude and adjusted hazard ratio [HR] for several baseline variables). Secondary outcome was the correlation between the overall incidence of HCC and statin administration expressed in terms of the odds ratio (OR), again both crude and adjusted.

## 2. Results

### 2.1. Characteristics of Included Studies

As shown in Figure 1, out of 5562 studies initially identified, after preliminary exclusion of manuscripts not fulfilling inclusion criteria, 95 potentially relevant articles were examined. Among these studies, 23 overlap series and 47 studies not reporting OR or HR (or data useful for their calculation) were further excluded.

Finally, 25 studies reporting 21,576 cases of HCC in 1,925,964 patients were included in the meta-analysis [8,9,10,11,12,13,14,15,16,17,18,19,20,21,22,23,24,25,26,27,28,29,30,31,32].

The main characteristics of the included studies are reported in Table 1.

The recruitment period ranged from 1988 to 2018. Nine studies [8,10,12,13,14,18,20,22] were retrospective case–control, twelve were cohort studies [9,11,15,16,17,19,23,24,26,27,29,30], one study was a randomized controlled trial (RCT) [28], and two studies that were included as RCTs represented individual patient data analysis of patients enrolled in prospective controlled trials of cholesterol in heart disease [31,32]. The study by Tran et al. [25] included two different stages, a nested case–control and a prospective cohort drawn from two different populations, hence they were analyzed separately.

Five studies [10,11,20,26,27] were conducted in the same population (Taiwan National Health Insurance and Research Database) but they reported different outcomes or data from different subgroups, therefore were considered separately in different analyses. Nine studies were conducted in Asia [10,11,15,16,18,20,26,27,28], and the remaining studies were conducted in western countries. The two aforementioned individual patient data analyses concerned multicenter RCTs [31,32].

Baseline clinical and demographical characteristics were well balanced between statin users and the control group. Variables adjusted for were mainly age, sex, and etiology of the underlying liver disease; other comorbidities; and the use of other medications.

Quality was deemed mainly high with five observational studies assessed as low-quality articles [13,16,19,23,29].

Details on methodological characteristics and quality of included articles are shown in Appendix A.

### 2.2. Risk of HCC

As reported in Figure 2A, overall crude OR for HCC incidence was 0.59 (95% CI: 0.47–0.74), thus highlighting a significant protective role of statins against HCC occurrence (*p* < 0.001). Of note, high evidence of heterogeneity was observed (I^2^ = 92%).

Adjusted analysis, considering the aforementioned baseline confounders, confirmed the anti-oncogenic effect of statins, with a reported adjusted OR (aOR) as high as 0.74 (95% CI: 0.70–0.78). Heterogeneity slightly decreased to 79% in adjusted analysis (Figure 2B).

Crude HR was reported only in two studies [17,29], confirming the above reported results in favor of statin use (HR: 0.42, 95% CI: 0.39–0.45), with moderate evidence of heterogeneity (I^2^=37%; Figure 3A). As described in Figure 3B, when adjusting for several clinical and demographical parameters, the HR slightly increased but remained under the significance threshold (adjusted [aHR]: 0.73, 95% CI: 0.69–0.76; I^2^=96%).

There was no evidence of publication bias (data not shown).

### 2.3. Subgroup Analysis

The aHR for HCC occurrence was confirmed as significantly in favor of statins in HBV patients (0.46, 95% CI: 0.36–0.60; I^2^ = 0%) while only a non-significant benefit was observed in HCV patients, although this result should be interpreted with caution due to the low number of studies and the high heterogeneity (Table 2).

Statins were proved to be effective in both diabetic and non-diabetic patients, while the magnitude of the chemopreventive effect was found to be linearly correlated to the dose, with an aHR of 0.51 (95% CI: 0.30–0.88) and 0.27 (95% CI: 0.11–0.67) in patients administered a cumulative defined daily dose (cDDD) below or beyond 365, respectively (Table 2).

Lipophilic statins (atorvastatin, lovastatin, and simvastatin) were associated with significantly reduced HCC (aHR: 0.49, 95% CI: 0.39–0.62, I^2^ = 19%) incidence while an association between hydrophilic statins (pravastatin, rosuvastatin, and fluvastatin) and reduced risk for HCC was not found (aHR: 0.73, 95% CI: 0 0.40–1.34) (Table 2).

Analysis conducted according to single agents found a significant beneficial effect only with atorvastatin (aHR: 0.43, 95% CI: 0.28–0.65, I^2^=17%), although further studies are warranted to provide definitive results in this regard.

### 2.4. Sensitivity Analysis

The results of several sensitivity analyses are reported in Table 3.

The findings of main analysis were confirmed in sensitivity analysis performed according to study quality (high versus low), design (RCT versus observational), and location (Asia versus western).

The only exception was the sub-analysis restricted to RCTs, where an OR around 1 was observed; of note, two out of three studies included as RCTs represented individual patient data analysis of patients enrolled in prospective controlled trials of cholesterol in heart disease (hence, non-cirrhotic) [31,32] where the incidence of HCC was very low.

Heterogeneity in the sensitivity analysis was mainly moderate.

## 3. Discussion

HCC is one of the most common cancer types and the leading cause of tumor-related deaths in cirrhotic patients [1].

Statins have been shown to consistently reduce liver fibrosis progression, mainly due to their immunomodulatory effects [33], to mitigate portal hypertension, and to upregulate transcription factors that exert vasoprotective effects in the liver and inhibit stellate cells, thus potentially further decreasing fibrosis [34]. On the other hand, the enthusiasm towards statins has been tempered by fears about their safety profile in cirrhotic patients because of the risk of dose-dependent hepatic injury [35].

A previous meta-analysis found a 37% decreased risk of HCC occurrence in statin users [7] but a specific analysis aiming to identify higher-risk settings was unfeasible due to the low number of included studies.

Through a meta-analysis of 25 studies, we made several key observations. First, we found a 26% decreased incidence of HCC in patients treated with statins, after adjustment for several variables. When considering a time-dependent outcome, such as HR, not influenced by the imbalance in follow-up length between the studies, we confirmed a 27% decreased incidence of HCC when adjusting for several clinical and demographical parameters.

Second, this effect was more pronounced and consistent in HBV patients (56% decrease in HCC incidence) and it was found to be linearly correlated to the dose, with a 73% decreased HCC risk in patients administered a cDDD beyond 365.

Third, as already observed in previous individual studies [24,27], lipophilic statins (atorvastatin, lovastatin, and simvastatin) were associated with significantly reduced HCC (51% compared to 27% in hydrophilic statin users). Among several agents tested, the more pronounced chemopreventive effect was observed with atorvastatin (57% reduction in HCC occurrence), although further studies are warranted to provide definitive results in this regard.

This preventive effect of statins is likely to be independent of its lipid-lowering effects, because lipid-lowering agents other than statins were not associated with reduction in the risk of HCC in previous reports [26,30].

The chemopreventive effect of statins in Asian populations, mainly affected by HBV hepatitis/cirrhosis, is well recognized and it was strongly confirmed in our analysis. In fact, HBV genome integration determines several DNA modifications and microdeletions that can target cancer-relevant genes, potentially providing hepatocytes with a growth advantage [36]. Statins, by inhibiting the mevalonate pathway, can prevent potential detrimental effects of these growth signaling proteins [37]. Statins also exert pro-apoptotic effects by activating several caspases and decreasing Bcl-2 [37]. Moreover, statins inhibit the activation of the proteasome pathway, limiting the breakdown of some molecules with growth-inhibitory effects, such as p21 and p27 [37].

On the other hand, the anti-oncogenic effect of statins in HCV patients is less evident, probably due to modification of metabolic syndrome, insulin-mediated cell proliferation, and obesity-associated inflammation [33,38]. A clear relationship between statin use and HCC incidence in HCV patients was not found in our meta-analysis, but this result might be due to the low number of studies specifically evaluating this subset of patients.

One of the novel findings in our study is the clear dose-dependent effect of statins in decreasing HCC occurrence, with 365 cDDDs as the cut-off to observe the highest preventive effects, as already reported in previous individual studies [28]. Several biological properties of statins, such as anti-angiogenetic or anti-fibrotic effects, were found to be strictly associated to the dose used in in vivo studies [39,40].

The greater chemoprotective effect of lipophobic statins, clearly outlined by our analysis, is likely due to greater lipid solubility and membrane permeability which enhance their pharmacological effects [41]. Further studies are needed to confirm these findings.

Of note, sensitivity analysis restricted to RCTs did not show a significant preventive benefit with statins although the result should be interpreted with caution due to the low number of RCTs included. It is evident that the limited number of RCTs was insufficient to detect a significant effect of statins, in particular considering that these RCTs were conducted in non-cirrhotic patients, hence with a very low HCC incidence.

There are some limitations to our study. First, the limited number of studies in many subgroups does not allow a strong comparison between statins users and non-users in several subsets of patients, in particular concerning single pharmacological agents or based on the use of specific antiviral treatments. Second, several comparisons were weakened by the high heterogeneity. We performed different sensitivity analyses that confirmed the main results and, noteworthy, the heterogeneity decreased when several subgroups were considered separately. To take into account the potential baseline confounders, we considered in our primary analysis the aHR. However, even if the heterogeneity decreased, it remained significant, probably none of the studies adjusted for the same confounders. Other eventual source of heterogeneity could be represented by the different populations enrolled in the included studies, with different risk of HCC occurrence and probably uneven screening campaigns in the different geographic areas. Finally, most of the included studies were retrospective series, hence prone to selection bias.

In conclusion, despite these weaknesses, our meta-analysis demonstrates the beneficial chemopreventive effect of statins against HCC occurrence. This effect is dose-dependent and more pronounced with lipophilic statins.

Further studies are warranted to confirm these results and to identify the exact setting where this anti-oncogenic effect could be enhanced.

## 4. Methods 

### 4.1. Inclusion and Exclusion Criteria

Only studies meeting the following criteria were included: (1) RCTs or observational studies recruiting >10 patients with clear exposure to statin therapy; (2) studies published in English; (3) articles reporting HCC occurrence; and (4) studies reporting OR, HR, or data useful for their calculation. Case reports, non-clinical studies, review articles, and animal models were excluded.

### 4.2. Search Strategy

Figure 1 reports the search strategy followed in the meta-analysis.

Bibliographic research was conducted on PubMed, EMBASE, Cochrane Library, and Google Scholar including all studies fulfilling inclusion criteria published until December 2019.

The search was conducted by two study investigators (AF and MAA) independently and keywords used were “HMG-CoA reductase inhibitor(s),” “statin(s),” “atorvastatin,” “fluvastatin,” “lovastatin,” “pravastatin,” “rosuvastatin,” or “simvastatin” combined with “liver cancer” or “neoplasm(s).”

Relevant reviews and meta-analyses on the use of statins and HCC occurrence were examined for potential suitable studies. Authors of included studies were contacted to obtain full text or further information when needed.

The quality of included studies was assessed by two authors independently (AF and MAA) according to the Cochrane Collaboration’s tool for assessing the risk of bias [42] for RCTs and the Newcastle–Ottawa scale [43] for non-randomized studies. Disagreements were solved by discussion and following a third opinion (RS).

### 4.3. Statistical Analysis

Primary endpoint of the current meta-analysis was the comparison of HCC occurrence between statin users and non-statin users. Data of HCC occurrence were compared through a random-effects model based on the DerSimonian and Laird test, and summary estimates were expressed in terms of both HR and OR along with their relevant 95% CIs.

To partially obviate the bias due to the different follow-up length among the studies and, within each study, between the two treatment arms and to consider not only the number of events but also their timing and the follow-up of censored patients, HRs were considered in the analysis when reported in the included studies.

Two separate analyses were conducted for crude and adjusted summary estimates (both OR and HR) and aHR was considered the primary endpoint in the meta-analysis.

Chi-square and I² tests were used for across studies comparison of the percentage of variability attributable to heterogeneity beyond chance. *p* < 0.10 for chi-square test and I² <20% were interpreted as low-level heterogeneity.

Probability of publication bias was assessed using funnel plots and with Begg and Mazumdar’s test.

Sensitivity analysis was conducted according to the quality of included studies (high versus low), location of the studies (Asia versus western), and study design (RCT versus observational).

A subgroup analysis based on several statin molecule and class (lipophilic versus hydrophilic), etiology of liver disease, presence of diabetes, and cumulative defined daily dose (cDDD: ≤365 versus >365) was performed.

All statistical analyses were conducted using RevMan 5.3 software (the Cochrane Collaboration, Oxford, UK). For all calculations a two-tailed p-value of <0.05 was considered statistically significant.

## 5. Conclusions

Our meta-analysis demonstrates the beneficial chemopreventive effect of statins against HCC occurrence. This effect is dose-dependent and more pronounced with lipophilic statins. Further studies are warranted to confirm these results and to identify the exact setting where this anti-oncogenic effect could be enhanced.

## Figures and Tables

**Figure 1 cancers-12-00874-f001:**
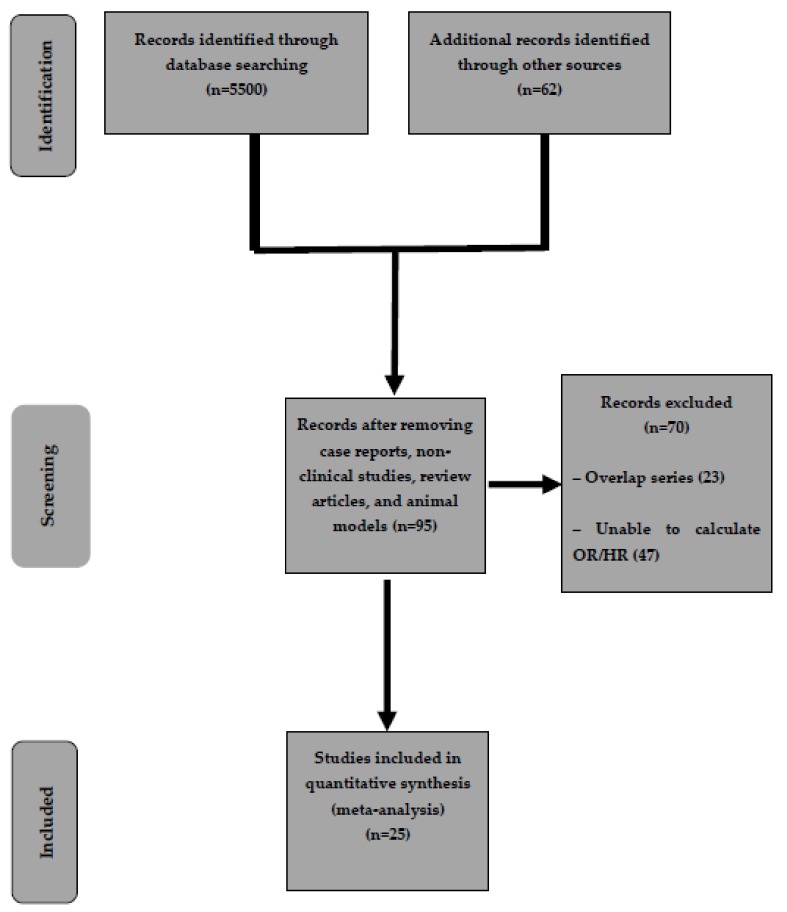
Flow chart of the search strategy conducted in this meta-analysis.

**Figure 2 cancers-12-00874-f002:**
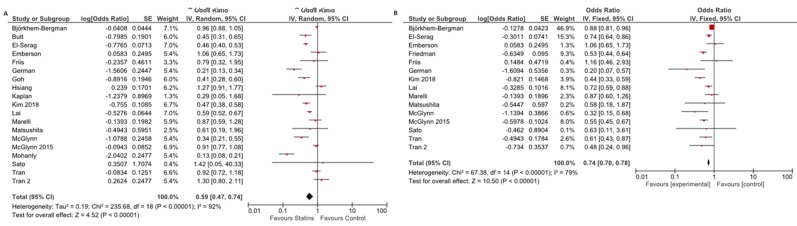
Odds ratio for HCC occurrence in the comparison between statin users and non-users: (**A**) crude odds ratio; and (**B**) adjusted odds ratio.

**Figure 3 cancers-12-00874-f003:**
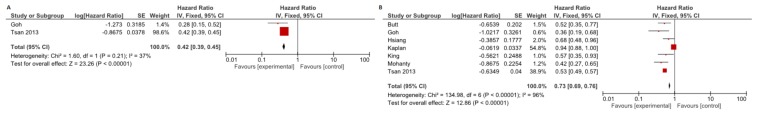
Hazard ratio for HCC occurrence in the comparison between statin users and non-users. (**A**) Crude HR. (**B**) Adjusted HR.

**Table 1 cancers-12-00874-t001:** Characteristics of included studies.

Study, Year	Design, Period	Country	Sample Size	Age	Men, *n* (%)	Liver Disease Etiology (HBV/HCV/alcohol/NASH)	Follow-Up Period	Statin Use Period or Dose	Outcome Measure	Variables Adjusted for
Björkhem-Bergman, 2014 [8]	Case–control, July 2006 to December 2010	Swedish Cancer Register	HCC group: 3994 patients (of which 687 statin users) Control (non-HCC), 19970 patients (of which 3598 statin users)	Mean age NR	52%	NR	4 years	At least 9 months	OR, aOR	Age, sex, diabetes, education, other drugs, liver disease etiology
ERCHIVES: Butt, 2015 [9]	Retrospective cohort, between 2002 and 2013	USA	Statin users: 3347 Non-users: 3901	Statin users: 53 (Non-statin users: 52	Statin Users: 3226 (96.4%) Non-users: 3702 (94.9%)	HCV all patients	10 years	Mean (IQR) months: Statin users: 31.7 (13.3–58.5)	HR, aHR	Baseline FIB-4
Chang, 2017 [10]	Nested case–control (propensity score matching retrospective study), 1 January 2000 to 31 December 2013	Taiwan NHIRD database (Taiwan’s National Health Insurance)	Statin users: 675 Non-users: 675	56.5±11.2 57.5 (14.1)	Statin users: 492 (73%) Non-users: 476 (71%)	313 (46%)/146 (22%)/219 (32%), 292 (43%)/152 (23%)/231 (34%)	Statin users: 5.5 years (3.5) Non-users: 5.4 years (3.6)	Patient with cDDDs >28 were considered as statin user	aHR	Age, sex, diabetes, comorbidities, other drugs, liver disease etiology
Chen, 2014 [11]	Propensity score matching retrospective cohort, 1 January 2000 to 31 December 2008	Taiwan’s National Health Insurance (NHI) Research Database (NHIRD)	Statin users*:* 8861 Statin + metformin: 5152 Non-users: 53037	Mean NR	Statin users*:* 4869 (54.95%) Statin + metformin: 2650 (51.44%) Non-users: 30726 (57.93%)	All patients HBV	9 years	Patients who used statins for <28 cDDDs were defined as statin non-users	aHR	Age, sex, comorbidities, other drugs
El-Serag, 2009 [12]	Nested case–control, 1997–2002	Department of Veterans Affairs (VA) National Databases, USA	HCC group: 1303 patients of which 447 statin users Control (non-HCC): 5212 patients of which 2766 non-users	HCC group: 72 years Control (non-HCC): 72 years	HCC group: 1286 (99%) Control (non-HCC): 5144(99%)	HCC group: 25 (1.9%)/192 (14.7%)/215 (16.5%)/ Control (non-HCC): 11 (0.2%)/93 (1.8%)/60 (1.2%)	5 years	Statin use defined as >3 prescriptions	OR, aOR	Etiology of liver disease, cirrhosis, race, other drugs
Friedman, 2016 [13]	Case–control, 1 January 1996 to 30 June 2014	Kaiser Permanente Northern California, USA	HCC group: 2877 patients of which 701 statin users Control group (non-HCC): 142850 patients of which 44953 statin users	NR	NR	NR	18 years	NR	aOR	Liver disease etiology, comorbidities, other drugs, BMI
German, 2019 [14]	Case–control, 2002–2016	Wisconsin (USA)	HCC group: 34 patients of which 6 statin users Control group (non-HCC): 68 patients of which 34 statin users	HCC group: 68.9±11.4 Control group (non-HCC): 69.4±7.5	HCC group: 22 Control group (non-HCC): 44	NAFLD all patients	14 years	NR	OR, aOR	Age, sex, other drugs
Goh, 2019 [15]	Retrospective cohort, January 2008 to December 2012	Single institution in Seoul, Republic of Korea	Statin users: 713 Non-users: 7000	Statin users: 50 (44–56) Non-users: 47 (39–54)	Statin users: 482 (67.6%) Non-users: 4624 (66.1%)	HBV (all patients)	7.2 years (0.5–9.7)	cDDD >28 was considered as statin use	HR, aHR	Age, sex, liver cirrhosis, comorbidities, viral level, other drugs, liver function tests
Hsiang, 2015 [16]	Propensity score matching retrospective cohort, January 2000 to December 2012	Hospital Authority (HA) registry database, Hong Kong	Statin users: 1176 Non-users: 52337	Statin users: 58.7±12.4 Non-users: 58.9±12.9	NR	HBV (all patients)	Statin users: 1.6 years (0.7–3.9) Non-users: 2.6 years (1–5.1)	cDDD: 291.5	aHR	–
Kaplan, 2019 [17]	Propensity score matching retrospective cohort, 1 January 2008 through 30 June 2016	Veterans’ Health Administration	Statin users: 21921 Non-users: 51023	Statin users: 64 (60–69) Non-users: 63 (58–68)	Statin users: 21373 (97.5%) Non-users: 12602 (98%)	Statin users: NR/2457 (11.2%)/8471 (38.6%)/5158 (23.5%) Non-users: NR/2065 (14.9%)/4876 (35.2%)/2159 (15.6%)	Statin users: 900 days (478–1546) Non-users: 1970 days (1234–2736)	270 days (0–827)	aHR	Race, liver disease etiology, liver function tests, cirrhosis, comorbidities, BMI
Kim, 2018 [18]	Nested case–control study, 2002–2013	National Health Insurance Service Physical Health Examination in the Republicof Korea.	HCC group: 1642 patients of which 111 statin users Non-HCC group: 8210 patients of which 1047 statin users	HCC group: 61.8±9.2 Non-HCC group: 61.8±9.2	HCC group: 1372 (83.6%) Non-HCC group: 6860 (83.6%)	HCC group: 755 (46%)/NR/277 (16.9%)/NR Non-HCC group: 232 (2.8%)/NR/418 (5.1%)/NR	NR		OR, aOR	Comorbidities, cirrhosis, BMI, other drugs, household income level
King, 2013 [19]	Prospective cohort	USA	136178	NR	NR	NR	>20 years		aHR	Age, BMI, comorbidities, other drugs
Lai, 2013 [20]	Case–Control study, 2000–2009	Taiwan National Health Insurance program	HCC group: 3480 patients of which 255 statin users Non-HCC: 13920 patients of which 1635 statin users	HCC group: 62.7±13.4 Non-HCC: 62.2±13.7	HCC group: 2525 (72.6%) Non-HCC: 10100 (72.6%)	HCC group: 1295 (37.2%)/1005 (28.9%)/66 (1.90%)/72 (2.07%) Non-HCC: 424 (3.05%)/274 (1.97%)/75 (0.54%)/86 (0.62%)	9 years	HCC group: 16.7 months Non-HCC: 18.6 months	OR, aOR	Age, sex, comorbidities, cirrhosis, etiology of liver disease, other drugs
McGlynn, 2014 [21]	Nested case–control, between 1999 and 2010	Population of the Health Alliance Plan HMO of the Henry FordHealth System (HFHS), a single integrated health system. USA	HCC group: 94 patients of which 25 statin users Non-HCC group: 468 patients of which 233 statin users	Mean NR	HCC group: 70 (74.47%) Non-HCC group: 348 (74.36%)	HCC group: 1 (1.06%)/46 (48.94%)/24 (25.53%)/NR Non-HCC group: 1 (0.21%)/8 (1.71%)/4 (0.85%)/NR	NR	*≤2 years:* HCC group: 13 Control group: 105 *> 2 years use of statin:* HCC group: 12 Control: 128	OR, aOR	Race, etiology of liver disease, comorbidities
McGlynn, 2015 [22]	Nested case–control, 1988 and 2011	UK’s Clinical Practice Research Datalink (CPRD).	HCC group: 1195 patients of which 302 statin users Non-HCC group: 4640 patients of which 1242 statin users	HCC group: 97.2±12.1 Non-HCC group: 67±12.1	HCC group: 856 (71.6%) Non-HCC group: 3322 (71.6%)	HCC group: 74 (6.2%)/189 (15.8%)/170 (14.2%) Non-HCC group: 23/(0.5%)/189 (4%)/9 (0.2%)	NR	*Cumulative dose: <8120)* HCC: 168 (14.1%) Control: 642 (13.2%) *>(21 281 * HCC: 152 (12.7%) Control: 649 (14%)	OR, aOR	BMI, etiology of liver disease, comorbidities, other drugs used
Mohanty, 2016 [23]	Propensity score matching retrospective cohort, January 1996 through December 2009	Veteran Affairs Clinical Case Registry, which contains nationwide data from veterans infected with the HCV	Statin users: 685 Non-users: 685	Statin users: 56 (52–59) Non-users: 56 (52–60)	Statin users: 677 (98.8%) Non-users: 671 (97.9%)	All had HCV and compensated cirrhosis	NR	NR	HR	–
Simon, 2019 [24]	Propensity score matching cohort study, 2005–2013	Swedish registers	Statin users: 16668 Non-users: 8334	Statin users: 47.3±11 Non-users: 47.5±13.7	Statin users: 65.2% Non-users: 65.6%	Statin users: 1540 (23.5%)/5014/6554 (76.5%)/NR Non-users: 1953 (23.4%)/6381/76.6%	8 years	NR	aHR	Age, sex, duration of viral infection, cirrhosis, comorbidities, other drugs used
Tran, 2019 [25]	Nested case–control, 1999-2011	Scottish Primary Care Clinical Informatics Unit (PCCIU) database.	HCC group: 434 patients of which 111 statin users Non-HCC group: 2103 patients of which 571 statin users	Mean NR	HCC group: 292 (67.3%) Non-HCC group: 1412 (67.1%)	NR	NR	HCC group: 4.88 years (3.1–7.29) Non-HCC group: 4.83 years (3.1–7.24)	OR, aOR	Age, sex, obesity, comorbidities, other drugs used, alcohol
Tran, 2019 (II) [25]	Prospective cohort	UK Biobank	Statin users: 395301 Non-users: 76550	Mean NR	NR	NR	NR	NR	OR	Age, sex, body mass index, alcohol, comorbidities, other drugs used
Tsan, 2012 [27]	Retrospective cohort, 1997–2008	Taiwan National Health Insurance Research Database	Statin users: 2785 Non-users: 30628	Statin users: 34.7 (26.6–43.8) Non-users: 46.3 (38.9–55.3)	Statin users: 1590 (57.1%) Non-users: 17852 (58.3%)	All patients have HBV	NR	28–90 cDDD: 933 (33.5%) 91–356 cDDD: 1279 (45.9%) >365 cDDDs: 573 (20.6%)	HR, aHR	Age, sex, income, diabetes, and liver cirrhosis
Tsan, 2013 [26]	Retrospective cohort, 1 January 1999 to 31 December 2010	Taiwan National Health Insurance Research Database	Statin users: 35023 Non-users: 225841	Statin users: 53.9 (45.4–62.1) Non-users: 49.8 (38.9–62)	Statin users: 14973 (42.8%) Non-users: 113290 (50.2%)	All patients had HCV	Statin users: 12 years (12.0–12.0) Non-users: 12 years (10.9–12)	179.6 CDD (80.0–414.7)	HR, aHR	Age, sex, urbanization, income, liver cirrhosis, and diabetes
Sato, 2006 [28]	RCT 28 September 1991 and 31 March 1995	Japan	Statin users: 179 Non-users: 84	NR	NR	NR	NR	All patients used pravastatin	OR, aOR	–
Marelli, 2011 [29]	Retrospective cohort, propensity score matching, 1990–2009	General Electric Centricity electronic medical records database	Statin users: 45857 Non-users: 45857	Statin users: 64.2±10.44 Non-users: 64.19±9.45	Statin users: 23953 (52.23%) Non-users: 24106 (52.57%)	Viral 28 (0.06%)	Statin users: 8.43 years Non-users: 8.43 years	NR	OR, aOR	–
Friis, 2005 [30]	Population-based cohort study, 1989–2002	The Prescription Database of North Jutland County and the Danish Cancer Registry	Statin users: 12251 Non-users: 322503	Statin users: 60.7 Non-users: 53.9	6935 (57%) 707 (56%)	NR	3.3 years (0–14)	Number of statin prescriptions: 2–4: 2392 (20%) 5–9: 2516 (21%) 10–19: 3282 (27%) 20+: 4061 (33%)	aOR	Age, gender, other drugs used
Matsushita, 2010 [31]	Individual patient meta-analysis of RCT	Multicenter	Statin users: 7375 Non-users: 6349	Statin users: 57.9±8.3 Non-users: 57.1±8.7	Statin users: 47.4% Non-users: 49.5%	NR	5.3 years	All patients used pravastatin	OR, aOR	–
Emberson, 2012 [32]	Individual patient meta-analysis from RCTs	International	Statin users: 67258 Non-users: 67279	63	46675 (27%)	NR	4.9 years	NR	OR, aOR	–

Data are reported as mean (standard deviation or interquartile range) or absolute number (percentage). Abbreviations: aHR, Adjusted hazard ratio; aOR, Adjusted odds ratio; cDDD, Cumulative defined daily dose; HCC, Hepatocellular carcinoma; HCV: Hepatitis C virus; HR, Hazard ratio; OR, Odds ratio; NR, Not reported; RCT, Randomized controlled trial.

**Table 2 cancers-12-00874-t002:** Subgroup analysis for adjusted hazard ratio concerning hepatocellular carcinoma occurrence.

Variable	Subgroup	Studies (*n*)	Summary Estimate (95% CI)	Within-Group Heterogeneity (I^2^)
Etiology of liver disease	HBV	2	0.46 (0.36–0.60)	0%
	HCV	2	0.68 (0.30–1.55)	66%
Diabetes	Yes	5	0.52 (0.46–0.58)	0%
	No	4	0.43 (0.31–0.61)	58%
Cumulative defined daily dose	≤365	3	0.51 (0.30–0.88)	78%
	>365	3	0.27 (0.11–0.67)	81%
Molecule	Lipophilic	2	0.49 (0.39–0.62)	19%
	Hydrophilic	2	0.73 (0.40–1.34)	83%
	Simvastatin	2	0.69 (0.42–1.15)	55%
	Atorvastatin	2	0.43 (0.28–0.65)	17%
	Fluvastatin	2	1.02 (0.08–13.25)	83%
	Pravastatin	1	0.80 (0.46–1.39)	NA
	Rosuvastatin	2	0.53 (0.04–6.38)	86%

Abbreviation: CI, Confidence interval; HBV, Hepatitis B virus; HCC, Hepatocellular carcinoma; HCV, Hepatitis C virus.

**Table 3 cancers-12-00874-t003:** Sensitivity analysis of the main diagnostic outcome (odds ratio for hepatocellular carcinoma occurrence) performed based on: (a) study design (observational versus RCT); (b) study location (Asia versus western); and (c) study quality (high versus low).

Variable	Subgroup	Studies (*n*)	Summary Estimate (95% CI)	Within-Group Heterogeneity (I^2^)
Study design	Observational	16	0.52 (0.41–0.73)	87%
	RCT	3	0.98 (0.76–1.32)	45%
Study location	Asia	8	0.51 (0.43–0.65)	44%
	Western	8	0.59 (0.45–0.81)	34%
Study quality	High	13	0.54 (0.44–0.89)	39.4%
	Low	3	0.57 (0.41–0.98)	55%
Abbreviations: CI, Confidence Interval; RCT, Randomized-Controlled Trial				

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
