# Peer review of "Statin Use Decreases the Incidence of Hepatocellular Carcinoma: An Updated Meta-Analysis"

_cancers, 2020, doi:10.3390/cancers12040874_

Round 1
Reviewer 1 Report
The authors set out an update of a systematic review and meta-analysis published by one of the co-author in 2013 in Gastroenterology. They aimed to evaluate the effects of statins on the risk of hepatocellular carcinoma (HCC) and found out that statins reduce the incidence of HCC by 26%, that this effect is dose-dependent and more distinct in patients harbouring HBV infection, and that lipophilic statins appears more effective in this context.
Their effort should be commended because the authors combined more recent and broad literature. They successfully confirmed the validity of previous assumptions and put forth new pieces of evidence. The article has the potential to impact current clinical practice. Therefore, it should be accepted for publication and, nonetheless, I have some minor criticism to rise:
- Was the protocol for the new systematic review registered in PROSPERO or in any other international prospective register of systematic reviews?
- Why the authors selected only studies published in English? Is there any relevant literature published in other languages that may have been skipped?
- It catches the eye that the conclusions of the work are substantially based on retrospective studies. In the subgroup analysis per study design, RCTs did not provide conclusive evidence. It is correct that this effect stems from selection, and that a study sample bereft of cirrhotic patients is not likely to generate enough effect to be detected in terms of hepatocellular carcinoma prevention. Yet, the authors might want to elaborate on the fact that retrospective studies can produce biased evidence as well.
Author Response
Reviewer 1:
The authors set out an update of a systematic review and meta-analysis published by one of the co-author in 2013 in Gastroenterology. They aimed to evaluate the effects of statins on the risk of hepatocellular carcinoma (HCC) and found out that statins reduce the incidence of HCC by 26%, that this effect is dose-dependent and more distinct in patients harbouring HBV infection, and that lipophilic statins appears more effective in this context.
Their effort should be commended because the authors combined more recent and broad literature. They successfully confirmed the validity of previous assumptions and put forth new pieces of evidence. The article has the potential to impact current clinical practice. Therefore, it should be accepted for publication and, nonetheless, I have some minor criticism to rise:
1) Was the protocol for the new systematic review registered in PROSPERO or in any other international prospective register of systematic reviews?
RE: The protocol was not registered in PROSPERO nor in other registers. This practice, albeit suggested by many publishers, is not compulsory for systematic reviews and meta-analyses (unlike randomized trials). This is particularly true considering that this is an update of a previous meta-analysis, which was registered to PROSPERO.
2) Why the authors selected only studies published in English? Is there any relevant literature published in other languages that may have been skipped?
RE: No, there are no other relevant studies published in other languages. However, it is common use to specify if language restriction was applied.
3) It catches the eye that the conclusions of the work are substantially based on retrospective studies. In the subgroup analysis per study design, RCTs did not provide conclusive evidence. It is correct that this effect stems from selection, and that a study sample bereft of cirrhotic patients is not likely to generate enough effect to be detected in terms of hepatocellular carcinoma prevention. Yet, the authors might want to elaborate on the fact that retrospective studies can produce biased evidence as well.
RE: We thank the reviewer for the important suggestion. We have now commented in the Discussion (Page 15): “Of note, sensitivity analysis restricted to RCTs did not show a significant preventive benefit with statins although the result should be interpreted with caution due to the low number of RCTs included. It is evident that the limited number of RCTs was insufficient to detect a significant effect of statins, in particular considering that these RCTs were conducted in non-cirrhotic patients, hence with a very low HCC incidence.” We have also added among the limitations to the study (Page 15): “Finally, most of the included studies were retrospective series, hence prone to selection bias.”
Reviewer 2 Report
This study looks at the protective effect of statins on the devleopment of HCC. This meta-analysis demonstrates the beneficial chemopreventive effect of statins against HCC occurrence; the effect is dose-dependent and more pronounced with lipophilic statins. Some concerns may deserve the authors' attention.
- The protective effect of statin on HCC has been investigated in many studies; some are quite large with consistent conclusions (Fig. 2). The need of this meta-analysis is thus questionable.
- The quality of Fig. 1 is poor and difficult to read.
- In Table 2 regarding the etiology of HCC, many patients could have co-existing HBV+HCV, HCV+ DM or others. This effect cannot be demonstrated in this Table. Also, many patients with HBV- or HCV-related HCC could have taken specific anti-virals, and this information is also absent in the current analysis. These drawbacks may complicate the results and conclusions.
- The Discussion does not contain in-depth implications to elaborate the mechanism of the protective effect of statins.
Author Response
Reviewer 2:
This study looks at the protective effect of statins on the development of HCC. This meta-analysis demonstrates the beneficial chemopreventive effect of statins against HCC occurrence; the effect is dose-dependent and more pronounced with lipophilic statins. Some concerns may deserve the authors' attention.
1) The protective effect of statin on HCC has been investigated in many studies; some are quite large with consistent conclusions (Fig. 2). The need of this meta-analysis is thus questionable.
RE: We agree that many large studies investigated the protective effect of statins on HCC and this is exactly the reason why we decided to conduct this meta-analysis: to appraise and systematically synthetize the growing body of evidence in the field.
2) The quality of Fig. 1 is poor and difficult to read.
RE: We have amended the figure improving its quality.
3) In Table 2 regarding the etiology of HCC, many patients could have co-existing HBV+HCV, HCV+ DM or others. This effect cannot be demonstrated in this Table. Also, many patients with HBV- or HCV-related HCC could have taken specific anti-virals, and this information is also absent in the current analysis. These drawbacks may complicate the results and conclusions.
RE: We considered an analysis only in patients with a single etiology in the subgroup, so studies with a considerable proportion of patients with co-existing infections were excluded from the subgroup analysis.
Due to the low number of studies in each viral subgroup, a further dichotomization according to the antiviral used was not feasible. Following the important reviewer’s suggestion, this was listed among the limitations to the study.
4) The Discussion does not contain in-depth implications to elaborate the mechanism of the protective effect of statins.
RE: We have commented in the Discussion (Page 15): “Statins, by inhibiting the mevalonate pathway, can prevent potential detrimental effects of these growth signaling proteins [37]. Statins also exert pro-apoptotic effects by activating several caspases and decreasing Bcl-2 [37]. Moreover, statins inhibit the activation of the proteasome pathway, limiting the breakdown of some molecules with growth-inhibitory effects, such as p21 and p27 [37].”
Reviewer 3 Report
In this manuscript, the authors presented the beneficial chemopreventive effect of statins against HCC occurrence by meta-analysis. The subgroup analysis also revealed that lipophilic statins were associated with significantly reduced HCC incidence, and that favorable effect was found in HBV patients. However, Kaplan DE already published the paper about effects of hypercholesterolemia and statin exposure on survival in a large national cohort of patients with LC. Singh S also explained that statins are associated with a reduced risk of HCC. This preventive effect is not novel, or interesting.
This manuscript was written by meta-analysis, and each study had quite different background. There are many patients with HBV infection in Asian country, and also many patients with NASH in Western countries. The reason why favorable effect was found in HBV patients might be the racial differences. This fact should be examined more precisely and carefully.
Author Response
Reviewer 3:
1) In this manuscript, the authors presented the beneficial chemopreventive effect of statins against HCC occurrence by meta-analysis. The subgroup analysis also revealed that lipophilic statins were associated with significantly reduced HCC incidence, and that favorable effect was found in HBV patients. However, Kaplan DE already published the paper about effects of hypercholesterolemia and statin exposure on survival in a large national cohort of patients with LC. Singh S also explained that statins are associated with a reduced risk of HCC. This preventive effect is not novel, or interesting.
RE: As already commented in response to the previous reviewer, the aim of this meta-analysis was to systematically review and synthetize the great body of evidence in the field, analysing several studies such as that of Kaplan et al. The meta-analysis by Singh et al was published 7 years ago and since then several new reports were published; hence the need of an updated meta-analysis. The current manuscript included more than double the number of studies compared with the previous meta-analysis by Singh, and this allowed us to perform more subgroup analyses that were not feasible in the previous report.
2) This manuscript was written by meta-analysis, and each study had quite different background. There are many patients with HBV infection in Asian country, and also many patients with NASH in Western countries. The reason why favorable effect was found in HBV patients might be the racial differences. This fact should be examined more precisely and carefully.
RE: We don’t believe that racial differences may explain the different chemo-preventive effect. In fact, sensitivity analysis according to study location (Table 3) showed similar results both in studies conducted in Asia (odds ratio 0.51, 0.43–0.65) and in western countries (OR: 0.59, 0.45–0.81). We proposed our explanation to this observation in the discussion: “HBV genome integration determines several DNA modifications and microdeletions that can target cancer-relevant genes potentially providing hepatocytes with a growth advantage [36]. Statins, by inhibiting the mevalonate pathway, can prevent potential detrimental effects of these growth signaling proteins [37]………. On the other hand, it is less evident the anti-oncogenic effect of statins in HCV patients, probably due to modification of metabolic syndrome, insulin-mediated cell proliferation, and obesity-associated inflammation [33,38]. A clear relationship between statin use and HCC incidence in HCV patients was not found in our meta-analysis, but this result might probably be due to the low number of studies specifically evaluating this subset of patients”.
Round 2
Reviewer 2 Report
The revised version is much improved.
Reviewer 3 Report
OK,